# Development and Validation of the ‘Self-Efficacy in Dealing with Self-Harm Questionnaire’ (SEDSHQ)

**DOI:** 10.3390/ijerph20010788

**Published:** 2022-12-31

**Authors:** Nienke Kool-Goudzwaard, Stasja Draisma, Jaap van der Bijl, Bauke Koekkoek, Ad Kerkhof, Berno van Meijel

**Affiliations:** 1Parnassia Psychiatric Institute, Parnassia Academy, 2552DH The Hague, The Netherlands; 2Department of Health, Sports & Welfare, Research Group Mental Health Nursing, Inholland University of Applied Sciences, 1081HV Amsterdam, The Netherlands; 3Department on Aging, Netherlands Institute of Mental Health and Addiction (Trimbos Institute), P.O. Box 725, 3500AS Utrecht, The Netherlands; 4Research Group Social Psychiatry and Mental Health Nursing, HAN University of Applied Sciences, 6503GL Nijmegen, The Netherlands; 5Department of Clinical Psychology, Amsterdam Public Health Research Institute, Vrije Universiteit, 1081HV Amsterdam, The Netherlands; 6Department of Psychiatry, Amsterdam Public Health Research Institute, Amsterdam UMC (VUmc), 1081HZ Amsterdam, The Netherlands

**Keywords:** self-harm, self-efficacy, validation study

## Abstract

Clinicians find it challenging to engage with patients who engage in self-harm. Improving the self-efficacy of professionals who treat self-harm patients may be an important step toward accomplishing better treatment of self-harm. However, there is no instrument available that assesses the self-efficacy of clinicians dealing with self-harm. The aim of this study is to describe the development and validation of the Self-Efficacy in Dealing with Self-Harm Questionnaire (SEDSHQ). This study tests the questionnaire’s feasibility, test-retest reliability, internal consistency, content validity, construct validity (factor analysis and convergent validity) and sensitivity to change. The Self-Efficacy in Dealing with Self-Harm Questionnaire is a 27-item instrument which has a 3-factor structure, as found in confirmatory factor analysis. Testing revealed high content validity, significant correlation with a subscale of the Attitude Towards Deliberate Self-Harm Questionnaire (ADSHQ), satisfactory test-retest correlation and a Cronbach’s alpha of 0.95. Additionally, the questionnaire was able to measure significant changes after an intervention took place, indicating sensitivity to change. We conclude that the present study indicates that the Self-Efficacy in Dealing with Self-Harm Questionnaire is a valid and reliable instrument for assessing the level of self-efficacy in response to self-harm.

## 1. Introduction

Self-harm is a major problem in society and healthcare. Approximately 4% of the general population engages in self-harm [1,2,3]. In the psychiatric population, the frequency is much higher. Thirty-three percent of adult patients and up to 60% of adolescent inpatients report a history of self-harm [4,5,6,7,8]. As a consequence, healthcare providers in psychiatry often encounter patients who engage in self-harm. Many clinicians feel inadequate and incompetent in dealing with patients who practice self-harm [9,10,11,12,13]. They find it difficult to engage with self-harm patients and feel they lack the skills to understand and treat self-harm [14]. Although some of them find it rewarding to take care of patients who practice self-harm, many of them find it difficult to do so due to the many feelings that may rise [15] and a lack of knowledge [16]. When patients keep harming themselves repeatedly, staff experience more feelings of frustration, powerlessness and insecurity [16,17]. Patients also perceive a lack of staff knowledge and skills with regard to self-harm [18]. In daily clinical practice, these perceptions may lead to miscommunication between staff and patients, resulting in a vicious circle of increasing frustration, powerlessness and misunderstanding in both groups [19]. As a result, the risk of patient self-harm may increase further. At the same time, the negative experiences of staff members when encountering and treating patients who practice self-harm may lead to diminished self-efficacy in their clinical work [19]. Improving the self-efficacy of healthcare providers may be an important step toward accomplishing better treatment of self-harm.

Self-efficacy can be defined as a person’s belief in his or her capability to make use of certain skills based on his or her judgment in executing those skills [20]. As self-efficacy is related to a specific task and situation [21], the measurement of self-efficacy is preferably also task- and context-specific. Consequently, instruments must focus on self-efficacy in relation to the specific concept involved, which in this study is self-harm, defined as any intentional, direct and indirect harming of body tissue with a non-fatal outcome [1]. No self-efficacy instruments measuring healthcare providers’ behaviors in dealing with self-harm were found in the literature. The closest tool for this concept was the subscale of ‘Dealing effectively with self-harm patients’ (referring to the interaction with patients who self-harm) from the Attitude Towards Deliberate Self-Harm Questionnaire (ADSHQ) [22]. However, this subscale cannot be regarded as a sufficiently validated scale measuring the concept of self-efficacy to its full extent. To measure professionals’ self-efficacy in treating patients who practice self-harm, we developed the Self-Efficacy in Dealing with Self-Harm Questionnaire (SEDSHQ). This article describes the development and validation of the SEDSHQ.

## 2. Materials and Methods

### 2.1. Design

The authors conducted a methodological study that involved developing the SEDSHQ and testing its psychometric properties: feasibility, test-retest reliability (as a necessary condition for validity), internal consistency, construct validity and content validity. As part of construct validity, we performed confirmative factor analysis (CFA) to establish an interpretable factor structure and determined the convergent validity. The SEDSHQ’s sensitivity to measuring change in functioning over time was also assessed.

### 2.2. Development of the Instrument

The item pool of the first draft of the questionnaire consisted of 46 items based on the literature about self-harm and self-efficacy. The literature about communication (with the patient, family, friends and colleagues) in specific situations concerning self-harm, identification of stress factors and (the recognition of) early signs of imminent self-harm, as well as preventive and curative interventions carried out in specific situations related to self-harm, was used to construct the items [19]. As self-efficacy is concerned with perceived capability, the items were phrased in terms of able to do instead of will do [21].

To test content validity, the questionnaire was sent to eight experts in self-harm: seven of them in the field of mental healthcare and one person with personal experience. For each item, they indicated whether it was 1 (‘irrelevant and should be deleted’) 2 (‘relevance is unclear because the meaning is unclear’), 3 (‘relevant but in need of minor adjustment’) or 4 (‘relevant and clear formulation’). As a result, 12 items were deleted because they were indicated by the experts to be irrelevant, too specific or redundant. Some adjustments took place, partly to achieve unequivocal wordings and partly to make the items more precise. Some examples include the item ‘I think I am able to calm patients, for example by holding their hands or by embracing them’, which was deleted because it was too specific and already covered by the item ‘Regarding patients who self-harm, I think I am able to encourage them when they are desperate or sad’. The item ‘Regarding patients who self-harm, I think I am able to encounter them positively (without prejudices)’ was revised to read ‘Regarding patients who self-harm, I think I am able to encounter them without prejudices’. We reasoned that the word ‘positively’ concerned a non-specific judgement that should be avoided in questionnaires. The item ‘I think I am able to help patients understand that their self-harming behavior is mainly a way to reduce stress’ was revised to read ‘I think I am able to search for the function of self-harm with the patient’, because reducing stress is not the only function of self-harm. The final questionnaire consisted of 34 items, which are presented in Table 1.

As the final step in the development process, seven mental health nurses were asked to assess the feasibility of the final questionnaire. They were asked whether they thought the questionnaire was understandable and easy to complete, and they confirmed that it was. 

A four-point Likert scale was used to score these statements, ranging from ‘probably not’ to ‘definitely yes’. The choice for a four-point Likert scale from ‘probably not’ to ‘definitely yes’—a so-called asymmetric answering scale—was due to people tending to overrate their self-efficacy; they would sooner think they can instead of cannot accomplish something (Dunning–Kruger effect). Therefore, in self-efficacy scores, the ceiling effect (a positive skew to the favorite end) is often seen, which makes it difficult to detect changes in self-efficacy due to self-efficacy enhancing interventions. An asymmetric or unbalanced answering scale can decrease this ceiling effect [23]. The total score ranged from 34 to 136, with a higher score indicating a higher level of perceived self-efficacy.

### 2.3. Participants

The study sample consisted of healthcare providers at eight mental healthcare settings and one forensic-psychiatric hospital who took part in a training program aimed at improving professionals’ attitudes and behavior toward patients who practice self-harm [24]. All trainees (n = 360) were asked to complete the questionnaire at two points in time: pre-test (two weeks before the training) and post-test (four weeks after the training). Because this study involved clinicians only, review and approval by the ethics committee was not necessary under Dutch legislation.

### 2.4. Data Collection

The questionnaires were sent to the participants in the training program by e-mail or by post. Two hundred seventy participants completed the the pre-test measurement of this intervention study, and 174 also completed the post-test measurements. For validation of the SEDSHQ, we used the pre-test measurement of this intervention study. To measure its sensitivity to change, we used the post-test measurements as well. In addition, for the test-retest measurements, a subgroup of the study sample was asked to complete the questionnaire a second time before the start of training at a 10 week interval. This subgroup consisted of 78 participants from 3 psychiatric hospitals.

Each participant received detailed information about the study, the questionnaire, and instructions on how to complete it. The participants’ anonymity was guaranteed, and their returning the questionnaires served as consent for participation. We also collected the participants’ background variables (i.e., gender, age, years of employment in psychiatry, education, experience with self-harm (professional and private) and whether the participant had had previous training concerning self-harm).

A reminder was e-mailed to the participants in order to improve the response rate. The data collection took place during the training program, which lasted from 2009 to 2011. 

### 2.5. Data Analysis

To measure the test-retest reliability, the intraclass correlation coefficient (ICC) was assessed based on the two-way random effects model, with 0.70 as a recommended minimum standard [25]. The purpose of this test is to establish the reproducibility of the SEDSHQ, or the degree to which repeated measurements in stable persons without an intervention provide similar answers [26]. The instrument reliability (internal consistency) was also determined using Cronbach’s alpha, with 0.70 as an acceptable value [27].

To establish an interpretable factor structure of the SEDSHQ, confirmatory factor analysis was conducted. Two goals were set for CFA: to uncover the underlying structure of the items in terms of the number of interpretable and plausible factors and to reduce the number of items in the instrument. A parallel analysis (with a scree plot) was conducted to explore the number of interpretable factors for the CFA. The obtained sample was too small to perform both exploratory factor analysis (EFA) and CFA on different parts of the sample.

Multicollinearity was investigated in the correlation matrix. In pairs of items with correlation above 0.8, one item was removed. Items were included if they had factor loadings of more than 0.4 [27]. Finally, items that loaded on more than one factor were deleted.

In the CFA, the following goodness-of-fit measures were used, and the acceptable values were specified according to [28]: comparative fit index (CFI) with a value >0.90, Tucker–Lewis index (TLI) values >0.90, root mean square error of approximation (RMSEA) values <0.06 and standardized root mean square residual (SRMR) values <0.08.

Convergent validity, which is another approach to testing the construct validity of a scale, was determined by comparing the total score of the SEDSHQ with the subscale ‘Dealing effectively with self-harm patients’ of the ADSHQ [22] using Pearson’s correlation [24]. The ADSHQ is a validated questionnaire which measures the attitudes of healthcare workers toward patients who engage in self-harm [22].

To establish the SEDSHQ’s sensitivity to change, a paired sample *t*-test was performed along with the pre-test and post-test measurements within this research design.

Prior to analysis, missing values analysis was performed. Questionnaires that had more than 20% missing values were deleted (n = 23). In questionnaires with less than 20% missing values, the ‘case mean substitution’ method was used to replace the missing value. According to Shrive et al. [29], this method can be used when up to 30% of items are missing.

IBM SPSS Statistics for Windows, version 20.0 (Armonk, NY, USA: IBM Corp.) was used to perform most statistical analyses. For the factor analysis, the package LAVAAN in R was used.

## 3. Results

### 3.1. Participants 

A total of 270 questionnaires were completed, with a response rate of 75%. Twenty-one percent of the responders were men, and 79% were women. Their average age was 39 years (range: 19–63 years), and they had worked an average of 12 years in (forensic) psychiatry (range: 0.5–39 years). Most of the participants had professional experience with self-harm (93%), and 21% also had personal experience with self-harm (themselves, family or friends). Only 5% had had previous training concerning self-harm. Table 2 provides information about the professions of the participants.

### 3.2. Reliability

The ICC between the first and second test (at a 10 week interval) was 0.71, indicating the high stability of the questionnaire. Cronbach’s alpha reliability coefficient was calculated and was found to be high (0.95).

### 3.3. Factor Analysis

The scree plot (Figure 1) results with parallel analysis showed a sharp inflexion point for one factor yet also adequate support for three or four factors, as expressed in adjusted eigenvalues above zero. The three-factor model was the only one which turned out to be theoretically interpretable and plausible. A first discerned factor in this model concerned items denoting the ‘own skills’ of the professional to identify self-harm signals and respond adequately to such situations. The second factor can be interpreted as communication with the patient to maintain a positive relationship aimed at agreement with respect to the identification and handling of difficult situations. The last factor contained three items about how to involve patients’ systems of contact (family and friends).

Three steps were taken to remove redundant items. First, the correlation matrix was checked for pairs of items with correlations above 0.8, denoting multicollinearity. Three pairs of items did have higher correlations: the least important item—at-face-value judgment—was removed. Thus items 14, 21 and 29 were discarded. Item 13 (‘talk to patients about feelings of anger and/or aggression’) was preferred, and item 14 (‘talk to patients about feelings of guilt and shame’) was discarded with respect to talking about emotions. Item 22 (‘investigate with friends/family the signs of imminent self-harm in the patient’) was preferred over item 21 (‘investigate with friends/family sources of patient stress’), since signs of self-harm are more easily observable than sources of stress. Item 28 (‘talk to patients about how to diminish stress’) was preferred over item 29 (‘investigate with patients which actions could be effective in cases of imminent self-harm’), since it was shorter and had a comparable meaning.

Items were included only when they uniquely loaded more than a result of 0.4 on a dimension (see Table 3: Factor loadings SEDSHQ). Three items had insufficient loadings on a factor (<0.4): item 3 (‘encounter them without prejudices’), item 15 (‘discuss with colleagues my own feelings about self-harming behavior’) and item 33, which were removed. The concepts in these two items, ‘no prejudice’ and ‘discuss with colleagues’ clearly did not belong to any of the three factors. Item 33 (‘deliver unconditional support after self-harm’) also loaded less than a result of 0.4 on factor 2 and was removed. Finally, item 30 (‘agree with patients how to involve family and friends in the treatment’) loaded almost equally high for both ‘patient communication’ as well as ‘contact patient system’, so it was also removed.

The final version of the list consisted of 27 items, and it is presented in Table 3 (Final version of the SEDSHQ: items and factor loadings). Since the SEDSHQ instrument was new, we used the modification index of the model in CFA to unconstrain some covariances between items in order to improve the values of the fit measures.

The following values for the fit measures of the three factor models were found: CFI = 0.907, TLI = 0.896, RMSEA = 0.071, CI [0.64–0.77] and SRMR = 0.057. These fit values indicate adequate fitting of the model. The Cronbach’s alpha values of the scales were the following: own skills = 0.87, patient communication = 0.93, contact patient system = 0.88 and total scale = 0.95.

### 3.4. Convergent Validity

Concerning the relationship between the total score of the SEDSHQ and the score of the ADSHQ subscale ‘Dealing effectively with self-harm patients’, a moderate, positive correlation was found between the two variables, where r = 0.52 (n = 257; *p* < 0.0005), indicating convergent validity.

### 3.5. Sensitivity to Change

The pre-test and post-test SEDSHQ scores of the professionals who participated in the training aimed at acquiring knowledge and skills regarding self-harm showed a significant increase, with a pre-test score of 92.59 (SD = 16.61) and a post-test score of 101.77 (SD = 15.73). This difference in pre-test and post-test scores was significant (t = −8.55; df = 173, *p* < 0.0001, r = 0.55) [20], indicating that the SEDSHQ is sensitive to changes.

## 4. Discussion

The aim of this study was to describe the development and psychometric properties of the Self-Efficacy in Dealing with Self-Harm Questionnaire (SEDSHQ). The results of this study indicate that the psychometric properties were within an acceptable range. It has high content validity, is stable over time when no intervention takes place and is able to measure changes after an intervention, indicating sensitivity to change.

The SEDSHQ had a moderate positive correlation with the ADSHQ subscale ‘Dealing effectively with self-harm patients’, indicating convergent validity. A three-factor solution was chosen covering the concepts of self, relation with patient and relation patient system in self-efficacy when treating patients who practice self-harm.

The questionnaire was developed to measure self-efficacy among healthcare professionals in mental healthcare, whether that be inpatient or outpatient care. The focus of the questionnaire is the direct interaction between healthcare providers and patients who practice self-harm and their families, and it refers to the necessary competencies to respond adequately to self-harm, thereby distinguishing the SEDSHQ between two other instruments measuring the attitudes of healthcare staff toward self-harm: the Self-Harm Antipathy Scale (SHAS) [30] and the Attitude Towards Deliberate Self-Harm Questionnaire (ADSHQ) [22]. The SHAS measures different aspects of empathy and antipathy toward self-harm, and the ADSHQ also measures cognitive and behavioral aspects regarding self-harm along with the attitude [31]. To raise awareness of the competencies, the level of functioning and the attitudes of individual professionals as well as a team as a whole, the SEDSHQ could be combined with (one of) the two attitude questionnaires. The acquired information about the level of functioning and attitudes can be used to offer specialized education to complement the deficits and strengthen the existing competencies and attitudes, thus improving clinical practice for patients who engage in self-harm.

A limitation of this study is that the sample consisted of professionals working in mental healthcare settings. Although this is the group of professionals who encounter patients who practice self-harm most frequently, other groups of professionals are confronted with self-harm as well. Further studies are required based on other samples, such as professionals working in general hospitals, general practitioners and school nurses. Another limitation is that the data were not collected recently. However, despite ongoing research regarding self-harm, no fundamental changes in theory, interventions or dealing with self-harm were encountered. Therefore, the items of the SEDSHQ formulated in the research period are still relevant. Finally, the sample size of this study was not large enough to allow application of exploratory and confirmative factor analysis on the subsamples. This would have been the most complete approach for studying construct validity. Further application and testing of the SEDSHQ questionnaire on different samples is recommended.

## 5. Conclusions

People who practice self-harm are entitled to appropriate, high-quality care. Measuring the self-efficacy of treatment staff when treating patients who engage in self-harm can reveal important information about their perceived capabilities concerning self-harm and whether or not they believe they are capable of using their skills. Based on their level of self-efficacy, decisions can be made regarding the need for additional training in self-harm. The SEDSHQ seems to be a valid and reliable instrument for assessing the level of self-efficacy. We recommend using the SEDSHQ in further studies and, as the validation of a scale is an ongoing process, continuing to examine its reliability and validity.

## Figures and Tables

**Figure 1 ijerph-20-00788-f001:**
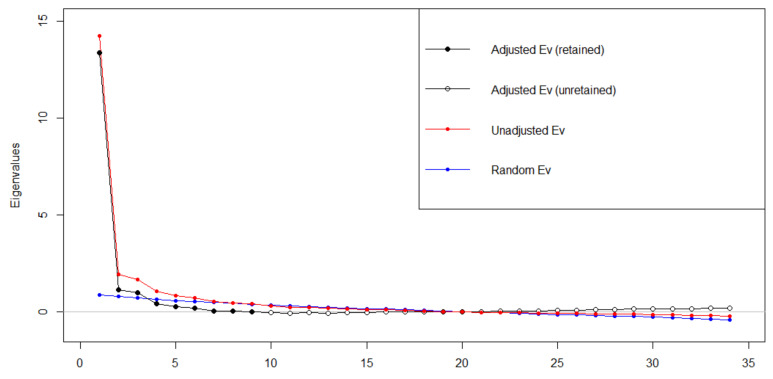
Scree plot for SEDSHQ.

**Table 1 ijerph-20-00788-t001:** Items of the Self-Efficacy in Dealing with Self-Harm Questionnaire (N = 270).

Regarding Patients Who Self-Harm, I Think I Am Able to …	Mean (Standard Deviation)
calm them in situations of stress or panic	2.7 (0.7)
2. encourage them when they are desperate or sad	2.7 (0.7)
3. encounter them without prejudices	2.8 (0.8)
4. estimate when patients need support and care	2.6 (0.6)
5. conclude how patients feel based on what they say	2.6 (0.6)
6. conclude how patients feel based on their behavior	2.6 (0.6)
7. make agreements with patients about communication (at what point, how long and in what way)	2.9 (0.7)
8. develop a confidential relationship with patients who challenge me	2.6 (0.7)
9. keep offering professional care, even if the self-harming behavior of patients continues for considerable time	2.7 (0.8)
10. discuss with patients which components of therapy they want to receive and at what pace	2.9 (0.8)
11. talk to patients about their self-harming behavior	3.0 (0.8)
12. investigate with patients the function of their self-harm	2.9 (0.8)
13. talk to patients about their anger and/or aggression	3.0 (0.7)
14. talk to patients about feelings of guilt and shame	2.9 (0.7)
15. discuss with colleagues my own feelings about self-harming behavior	3.2 (0.7)
16. make agreements with patients about how to handle subjects they do not want to talk about	2.6 (0.8)
17. recognize sources of stress in patients which can lead to self-harm	2.6 (0.7)
18. recognize signs of imminent self-harm in patients	2.5 (0.8)
19. investigate with patients’ possible sources of stress	2.9 (0.7)
20. investigate with patients which signs indicate imminent self-harm	2.9 (0.8)
21. investigate with friends/family sources of patient stress	2.4 (0.8)
22. investigate with friends/family the signs of imminent self-harm in the patient	2.4 (0.8)
23. investigate with patients their own sources of strength which help them prevent self-harm	2.7 (0.8)
24. recognize activities that patients use to handle increasing stress and imminent self-harm	2.6 (0.7)
25. deploy specific actions when there are signs of imminent self-harm	2.6 (0.8)
26. ascertain with patients immediate needs in cases of increasing stress and imminent self-harm	2.5 (0.8)
27. analyze with patients the relationship between events, thoughts, feelings and (self-harming) behavior	2.6 (0.8)
28. talk to patients about how to diminish stress	2.9 (0.7)
29. investigate with patients which actions could be effective in cases of imminent self-harm	2.8 (0.7)
30. agree with patients how to involve family and friends in the treatment	2.6 (0.7)
31. investigate with patients and family who will act in what way to diminish stress or prevent self-harm	2.4 (0.8)
32. help friends/family make use of the agreed actions to diminish stress or to prevent self-harm	2.3 (0.7)
33. deliver unconditional support after self-harm	2.6 (0.8)
34. investigate and prepare with patients the desired after-care actions	2.6 (0.8)

**Table 2 ijerph-20-00788-t002:** Professional backgrounds of the sample.

Education	Number	%
Certified nurse assistant	3	1.1
Registered nurse	132	48.9
Social worker	38	14.1
(Clinical) psychologist or psychotherapist	14	5.2
Psychiatrist	1	0.4
Occupational therapist	16	5.9
Other (e.g., trainer or manager)	7	2.6
Unknown	59	21.8
Total	270	100

**Table 3 ijerph-20-00788-t003:** Final version of the SEDSHQ: items and factor loadings.

	Factors			
Item *(Regarding Patients Who Self-Harm) I Think I Am Able to…*	Own Skills	Patient Commu-nication	Contact Patient System	Final Item Number
1. calm them in situations of stress or panic	0.473			1
2. encourage them when they are desperate or sad	0.494			2
4. estimate when patients need support and care	0.580			3
5. conclude how patients feel based on what they say	0.527			4
6. conclude how patients feel based on their behavior	0.426			5
17. recognize sources of stress in patients which can lead to self-harm	0.748			6
18. recognize signs of imminent self-harm in patients	0.670			7
24. recognize activities that patients use to handle increasing stress and imminent self-harm	0.782			8
25. deploy specific actions when there are signs of imminent self-harm	0.737			9
7. make agreements with patients about communication (at what point, how long and in what way)		0.622		10
8. develop a confidential relationship with patients who challenge me		0.515		11
9. keep offering professional care, even if the self-harming behavior of patients continues for considerable time		0.465		12
10. discuss with patients which components of therapy they want to receive and at which pace		0.572		13
11. talk to patients about their self-harming behavior		0.672		14
12. investigate with patients the function of their self-harm		0.704		15
13. talk to patients about their anger and/or aggression		0.702		16
16. make agreements with patients about how to handle subjects they do not want to talk about		0.649		17
19. investigate with patients possible sources of stress		0.835		18
20. investigate with patients which signs indicate imminent self-harm		0.827		19
23. investigate with patients their own sources of strength which help them prevent self-harm		0.757		20
26. ascertain with patients immediate needs in cases of increasing stress and imminent self-harm		0.720		21
27. analyze with patients the relation between events, thoughts, feelings and (self-harming) behavior		0.761		22
28. talk to patients about how to diminish stress		0.769		23
34. investigate and prepare with patients the desired after-care actions		0.706		24
22. investigate with friends/family signs of imminent self-harm in the patient			0.786	25
31. investigate with patients and family who will act in what way to diminish stress or prevent self-harm			0.865	26
32. help friends/family make use of the agreed actions to diminish stress or to prevent self-harm			0.876	27

Mean total scale = 72.3; standard deviation = 13.2. Mean scale own skills = 23.5; standard deviation = 4.3. Mean scale patient communication = 41.7; standard deviation = 8.2. Mean scale contact patient system = 7.1; standard deviation = 2.1.

## Data Availability

The data presented in this study are available on request from the corresponding author.

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
