# Peer review of "Development and Validation of the ‘Self-Efficacy in Dealing with Self-Harm Questionnaire’ (SEDSHQ)"

_ijerph, 2022, doi:10.3390/ijerph20010788_

Round 1

Reviewer 1 Report

In this manuscript, the authors describe the development and validation of a self-report measure, called the Self-Efficacy in Dealing with Self-Harm Questionnaire (SEDSHQ). A validated measure of self-efficacy in treating self-harm can be useful for evaluating outcomes for mental health professionals received trainings on self-harm. Concerns are noted below:

 Concerns:

*The SEDSHQ items were developed based on literature focused on self-harm and self-efficacy and by having experts review the items. In the Methods section, the authors note that data collection took place from 2009 to 2011. Given this, is it safe to assume that the items were developed prior to 2009? If so, the authors are recommended to specify when they developed the items and to consider in the limitations section if the current items reflect the latest literature on self-harm and self-efficacy.

*The authors note that the items on the SEDSHQ are rated on a 4-point Likert scale from ‘probably not’ to ‘definitely yes’ (pg. 4). Why did the authors use ‘probably not’ as one of the anchor responses instead of ‘definitely not’? The authors are encouraged to list out the anchors for each rating. Relatedly, Bandura (2006) recommended that efficacy scales that use only a few scale points should be avoided because they are less sensitive and reliable. The authors are encouraged to provide justification for their current scale range and/or consider the limitations of such a rating scale. The chapter by Bandura (2006) may also be relevant to cite in the introduction and discussion.

-      Bandura, A. (2006). Guide for constructing self-efficacy scales. In F. Pajares & T. Urdan (Eds.), Self-efficacy beliefs of adolescents (pp. 307–338). Charlotte, NC: Information Age Publishing.

*In the Participants subsection, the authors state that the participants were asked to complete questionnaire at two time points (pre-test and post-test) during the training program. When did the pre-test and post-test assessments occur in relation to the training?

*Table 3 and 4 are referenced in the body of the manuscript but are missing from the submission. Please add in these two tables. Additionally, it is recommended that the authors report the means and standard deviations of the total score and subscales of the final SEDSHQ.

*In the Discussion, the sentence beginning on line 247 needs to be reworded. The sentence states that the SEDSHQ is distinguishable from the Self Harm Antipathy Scale (SHAS) and the ADHSQ. Is the SHAS conceptually similar to the SEDSHQ and/or the ADHSQ? If so, the authors should mention this in the Introduction as well, in the paragraph where they discuss the ADSHQ.

*The authors are encouraged to consider other limitations of the current study. Currently, only one limitation is listed.

Minor concerns:

*In the introduction, the authors describe how healthcare providers may encounter patients who self-harm. The authors are encouraged to specify “mental health” or “psychiatric” health care providers given that the measure was designed with mental health care experts and participants were professionals in a psychiatric setting.

*The sentence beginning with “Especially” on line 74 of page 2 is a bit confusing. The authors are encouraged to reword the sentence.

*There appears to be an extra sentence or two in the second paragraph of page 6 (lines 203-208).  

Reviewer 2 Report

Thank you for this well-written and detailed paper describing the development of a questionnaire to assess clinicians' self-efficacy in treating patients who self-harm. I was very interested to review this paper, however, there are methodological concerns in addition to contextual information that need to be addressed:

- In the Abstract and the Introduction sections, the authors state that clinicians find it difficult to engage with patients who self-harm. This is quite a general statement and the references to supporting literature mainly seem to be challenges experienced by nurses. I think this could be described more accurately (e.g. is it certain disciplines and what is it specifically that makes it difficult for some clinicians to engage with? Those with repeat self-harm or high-risk self-harm?). I would also suggest that such information be presented with greater sensitivity. If someone was reading this work who had experienced self-harm, I think the presentation of this information in its current format has the potential to be stigmatising. 

- I was unsure what the authors meant by when they referred to the scale as "Dealing with" self-harm. Is this treatment of patients or interacting with patients? I think this could be labelled more clearly.

- Related to the previous point, a definition of the construct of interest is not provided here. This is extremely important in scale development to ensure accurate and precise measurement of the construct of interest. 

- The development of the instrument is outlined in detailed which is great. I agree with and commend the authors for the detailed process followed in the development of the items for inclusion in the scale. However, the response format is not in line with the recommended format for a self-efficacy scale as outlined by Bandura. This should be a scale from 0-100 (or similar) where participants rate how confident they feel they can complete the specified tasks. A Likert type scale is not recommended and is more suitable for measures of attitude.

- While the authors obtained a large number of participants for the study which is to be commended, the majority of the sample were nurses. Given that there may be potential differences in how effectively clinicians from different disciplines feel they can treat patients with self-harm, I wondered if it was appropriate to include all participants in one group for the analysis.

- I think it would have been more appropriate to conduct EFA on the current dataset and CFA at a later point. Typically, these analyses would be done on different samples so I think a focus on EFA here would have been more appropriate.

- I would like to have seen the factor loadings of the scale items. The authors refer to Table 3 but I could not see it in the manuscript or supplementary material. 

- For a self-efficacy questionnaire, I don't think it's required or relevant to conduct test-retest reliability as there are many factors that may influence individuals self-efficacy other than a training intervention. Instead, I think it would be more relevant to focus more on content and construct validity.
